# High Complication Rate and High Percentage of Regressing Radiolucency in Magnesium Screw Fixation in 18 Consecutive Patients

**DOI:** 10.3390/jpm13020357

**Published:** 2023-02-17

**Authors:** David J. Haslhofer, Tobias Gotterbarm, Antonio Klasan

**Affiliations:** 1Department for Orthopedics and Traumatology, Med Campus III, Kepler University Hospital Linz, Krankenhausstrasse 9, 4020 Linz, Austria; 2Faculty of Medicine, Johannes Kepler University Linz, Altenbergerstrasse 69, 4040 Linz, Austria; 3Department for Orthopedics and Traumatology, AUVA Graz, Göstinger Straße 24, 8020 Graz, Austria

**Keywords:** magnesium-based screws, radiolucency, orthopedics and traumatology

## Abstract

(1) Background: Magnesium-based implants use has become a research focus in recent years. Radiolucent areas around inserted screws are still worrisome. The objective of this study was to investigate the first 18 patients treated using MAGNEZIX^®^ CS screws. (2) Methods: This retrospective case series included all 18 consecutive patients treated using MAGNEZIX^®^ CS screws at our Level-1 trauma center. Radiographs were taken at 3-, 6- and 9-month follow-ups. Osteolysis, radiolucency and material failure were assessed, as were infection and revision surgery. (3) Results: Most patients (61.1%) had surgery in the shoulder region. Radiolucency regressed from 55.6% at 3-month follow-ups to 11.1% at 9-month follow-ups. Material failure occurred in four patients (22.22%) and infection occurred in two patients, yielding a 33.33% complication rate. (4) Conclusion: MAGNEZIX^®^ CS screws demonstrated a high percentage of radiolucency that regressed and seems to be clinically irrelevant. The material failure rate and infection rate require further research.

## 1. Introduction

Biodegradable alternatives to conventional metal screws were first used clinically in the 1980s [1]. In the early years, four criteria for biodegradable implants were set by Speer and Warren [2]. They stated that (1) The bioabsorbable implant must have adequate initial fixation strength to coapt the soft tissues to bone; (2) The implant’s bioabsorption profile must enable it to retain satisfactory strength while the healing tissues are regaining mechanical integrity; (3) The implant must not bioabsorb too slowly or it will behave like its metal counterpart with breakage and migration; and (4) The implant must be made of materials that are completely safe: no toxicity, antigenicity, pyrogenicity, or carcinogenicity [2].

Throughout the years, some advantages and disadvantages of using metal screws have been noted. Short-term complications mentioned in the literature include material fractures during screw insertion, lack of implant stability and screw migration. In the medium term, bony changes, such as lyses or cysts, swelling and abscesses have been described. The advantages of using biodegradable screws are simpler revisions, lack of artifacts in imaging and less soft tissue destruction [3]. In addition, biodegradable screws, such as magnesium-based screws, provide an option to avoid implant removal concomitant with secondary surgery, and thus potentially decrease infection risks [4,5].

Magnesium has some potentially interesting biological properties. Magnesium and its alloys are described as osteoinductive because of their similarity to bone structure, and have therefore received attention as an alternative in recent years [6,7]. The first European approval of magnesium-based implants for application in human bodies occurred in 2013, and was provided for a compression screw, MAGNEZIX^®^ CS, produced by Syntellix (Syntellix AG, Hannover, Germany) [4,8]. The first clinical studies compared the fixation of chevron osteotomies using conventional screws and magnesium-based MAGNEZIX^®^ CS screws [8,9,10]. Over time, further studies concerning chevron osteotomies followed [11,12,13], as well as studies concerning their use in children and adolescents [14,15,16]. In the meantime, MAGNEZIX^®^ CS screws have been used for different indications in various anatomical regions (foot/ankle, knee, elbow and shoulder), fracture fixation, osteochondral refixations and osteotomies [14,17]. Published clinical studies presented predominantly safe usage and promising outcomes [11,14,16,17,18,19].

Most of these studies have one worrisome phenomenon in common—the development of radiolucent areas around the inserted implants— which have been found of little or no clinical relevance, occur regularly and often regress [11,14,17,19,20].

The aim of our study was to investigate the first 18 patients treated using MAGNEZIX^®^ CS screws in our Level-1 trauma center for all indications, with a focus on shoulder instability and fractures, at any age.

## 2. Materials and Methods

### 2.1. Ethics

The study was approved by the local Regional Ethical Committee (ECS 1265/2022).

### 2.2. Patients

In this retrospective case series, consecutive patients treated using MAGNEZIX^®^ CS screws in our Level-1 trauma center between October 2021 and February 2022 were included. All patients had at least a minimum 9-month postoperative radiograph (X-ray). No patients were excluded.

Data regarding 18 consecutive patients were extracted, as shown in Figure 1.

We collected data regarding age, gender, side of injury, body mass index (BMI), body region of injury and the implants used. We further collected post-operative data regarding pain (visual analogue scale (VAS)), radiolucency, material failure and infection (positive postoperative wound swab).

Patients were treated via surgical intervention due to various indications, including shoulder instability, fractures and osteochondral refixation.

Radiographic evaluation was performed based on postoperative X-ray images. In cases in which the question of sufficient bony healing arose, postoperative computer tomographic (CT) scans were used to evaluate material failure and radiolucency.

### 2.3. Methods

Follow-up examinations were performed 3 months, 6 months and 9 months postoperatively for each patient. At each of these controls, pain was assessed using a visual analogue scale. In addition, radiological images were assessed for osteolysis or radiolucency around the inserted osteosynthesis material/MAGNEZIX^®^ CS screws and for material failure of these screws. Furthermore, it was documented whether an infection was present.

The assessment of all radiographic images concerning osteolysis, radiolucency and material failure was performed by two attendings and one resident. Radiolucency was defined as distinct gray values of the bony structure around the inserted screws.

Infections were defined as at least one positive wound swab in our documentation system. Tissue samples were obtained from the wound and were analyzed using standardized procedures.

Material failure was defined as broken screws or stabilization failure of the inserted screws in terms of postoperative dislocation.

### 2.4. Screws

We used MAGNEZIX^®^ CS produced by Syntellix (Syntellix AG, Hannover, Germany) in all 18 patients. The number of screws used ranged from 1 to 4. Screw diameters ranged from 2.0 mm to 3.2 mm and screw length ranged from 18 mm to 34 mm.

### 2.5. Statistics

Statistical analysis was performed using IBM SPSS Statistics 28 (Chicago, IL, USA). Data analysis was carried out using standard methods of descriptive statistics.

## 3. Results

All 18 extracted patients were included in this study. Age ranged from 9 to 64 years with a mean age of 36.3 years. The mean body mass index (BMI) was 25.9 (minimum 18.3, maximum 37) (Table 1).

In 11 patients, surgery was performed in the shoulder region (61.1%). Three patients were treated due to an injury of the elbow (16.7%). MAGNEZIX^®^ CS screws were used in one patient for each of the forearm, foot, ankle and knee regions (5.6% each) (Figure 2).

### 3.1. Radiolucency

At 3-month follow-ups, 10 patients’ (55.6%) X-rays showed radiolucency. A significant decrease in radiolucency was seen in the further controls. At 6-month follow-ups, radiolucency was still visible in five patients (27.8%), and at 9-month follow-ups radiolucency was visible in only two patients (11.1%) (Table 1).

### 3.2. Material Failure

Material failure always presented through screw breakage. After 3 months, this was observed in three patients.

In an 81-year-old patient, four MAGNEZIX^®^ CS screws (2.7 mm diameter) were used for osteosynthesis in a glenoid fracture after anterior shoulder dislocation. Here, material failure was seen in three of the four screws (Figure 3). At the follow-up examinations (3, 6 and 9 months), the patient was symptom-free, except for minimal pain (VAS 1).

Another patient in whom material failure of the screws was observed underwent shoulder surgery. The 25-year-old patient was treated for recurrent shoulder dislocation using Latarjet surgery. Both MAGNEZIX^®^ CS screws (3.2 mm diameter) used in the surgery were found to be insufficient and fractured as of the 3-month radiographic follow-up. At the 3-month follow-up, the patient still reported mild pain (VAS 2). However, in the two further controls (6 and 9 months), the patient was symptom-free.

In both shoulder patients, however, bony stability was already present at their 3-month follow-ups, so that the material failure had no further effects.

The third patient in whom a material failure was suspected at the 3-month follow-up was a 12-year-old boy. He underwent refixation of an osteochondral fragment at the capitulum humeri. Two MAGNEZIX^®^ CS screws (2.0 mm diameter) were used. In this patient, too, osteochondral healing was already sufficient in the case of material failure, so that no further therapy was required. At the 6- and 9-month follow-ups, the patient reported discrete pain, with no complaints otherwise.

At the 9-month follow-up, screw fractures were evident in one additional patient. Here, 2 MAGNEZIX^®^ CS screws (3.2 mm diameter) were used in a Latarjet surgery indicated by recurrent shoulder dislocations in a 27-year-old patient. The patient had already received an arthroscopic Bankert repair two years prior to this procedure.

Radiolucency was still visible and not improved as of the 3-month follow-up, and at the following two as well. A build-through process of the coracoid bone block, initially fixed by MAGNEZIX^®^ CS screws, was not achieved. Dislocation and lysis of the fragment occurred even after 9 months. Clinically, the patient presented with pain during the controls, especially during external rotation (with a VAS of 3 at 3 months, a VAS of 6 at 6 months and a VAS of 4 at 9 months). Instability and effusion or swelling did not present at any follow-up. The patient reported no fear of re-dislocation; therefore, a new surgical intervention was not performed at the time, at the patient’s request.

### 3.3. Infection

Postoperatively, two patients showed wound infection including a positive wound swab.

A 63-year-old female patient who underwent surgery for a bimalleolar ankle fracture, and in whom two MAGNEZIX^®^ CS screws (2.7 mm diameter) were used for osteosynthesis on the medial malleolus, presented with wound dehiscence after removal of the positional screw on the lateral malleolus. This showed as a single positive wound swab for Staphylococcus aureus. With oral antibiotic therapy, this could be eradicated; no further surgical therapy was necessary when the wound dehiscence disappeared.

The second patient, a 46-year-old, underwent humeral plating for a subcapital humerus fracture including fixation of the greater tuberosity using a MAGNEZIX^®^ CS screw (2.7 mm diameter). Postoperatively, an early infection with Staphylococcus aureus colonization was observed, which resulted in several revision procedures using vacuum assisted closure (VAC) therapy (14 cycles) and weeks of antibiotic therapy. Finally, two months after the initial fracture treatment, a mesh-graft could be applied to an infection-free site. At the 6- and 9-month follow-ups, the patient showed no complaints except minimal pain (with a VAS of 2 at 6 months and a VAS of 1 at 9 months).

### 3.4. Re-Operation

In addition to the latter patient, another patient in our collective required further surgery. Osteosynthesis of the forearm fracture of a 15-year-old boy used a MAGNEZIX^®^ CS screw (2.0 mm diameter) in addition to several k-wires. Tendolysis was necessary four months after the original treatment due to postoperative extensor tendon adhesions of three fingers. This surgery proceeded without complications. Except for minimal pain three months after the first operation due to the tendon adhesions, the young patient did not indicate any complaints in the further follow-ups.

## 4. Discussion

In this study, we provided an overview of the care outcomes of the first 18 patients who were treated using MAGNEZIX^®^ CS screws at our clinic.

Self-dissolving materials, especially MAGNEZIX^®^ CS screws, have become increasingly important in recent years. The advantage of being able to avoid implant removal was the focus here. At our clinic, despite knowledge that radiolucency can occur, discussions about the clinical results occurred repeatedly during presentations of postoperative radiographs.

The first magnesium-based implants were limited by too-rapid degradation, which led to insufficient fixation and stabilization of the bone. In addition, due to magnesium corrosion, major hydrogen gas formation occurred [15,21,22,23]. In recent years, the MAGNEZIX^®^ CS screws’ corrosion was improved by balancing regeneration of bone structure and implant resorption [8,15,18,24,25].

In our study, in addition to the care of young patients, the use of screws in shoulder surgery was a particular focus (61.1%).

Compared with previous studies, it could be shown in our investigated patient population that a significant amount of radiolucency occurred in the early phases of follow-up (55.6%), but that this regressed during the course of follow-up (down to 11.1% at 9 months FU) [11,14,17,19,20].

In their patient population, Stürznickel et al. described a pin fracture after fixation of an osteochondral defect at the medial femoral condyle [14]. In our study, this was comparable to a screw fracture three months after fixation of an osteochondral fragment at the capitulum humeri.

However, material failure was mainly visible in our patients after use of MAGNEZIX^®^ CS screws in shoulder surgery (two Latarjet procedures, and one osteosynthesis for glenoid fracture).

There are no conclusive clinical studies regarding the use of bioabsorbable screws in shoulder surgery (for fracture treatment or treatment of instabilities) [26,27]. In a biomechanical study, Bockmann et al. compared steel screws, polylactic acid (PLLA) screws and magnesium screws used during Latarjet procedures [28]. They reported that all three types of screws withstood axial forces greater than 200 Newton (N) [28]. In the case of magnesium screws, failure mode was reached primarily due to material fracture [28]. Our shoulder cases showed a failure rate of one-third (three out of nine patients).

In comparison, a systematic review by Hurley et al. showed short term complication rates of 6–7% after a Latarjet procedure [29].

In particular, the safe use of MAGNEZIX^®^ CS screws in the treatment of shoulder instability (specifically Latarjet procedures) requires further investigation via clinical studies.

Regarding our two patients who suffered infections, there was no clear association with the use of MAGNEZIX^®^ CS screws. Additionally, in the literature, no difference was reported regarding infectivity compared to conventional screws [11,12].

In addition to its retrospective study design, the heterogeneous patient sample was a limitation of our study.

## 5. Conclusions

MAGNEZIX^®^ CS screws demonstrated a high percentage of radiolucency that regressed and seemed clinically irrelevant. Their material failure rate and infection rate require further research.

## Figures and Tables

**Figure 1 jpm-13-00357-f001:**
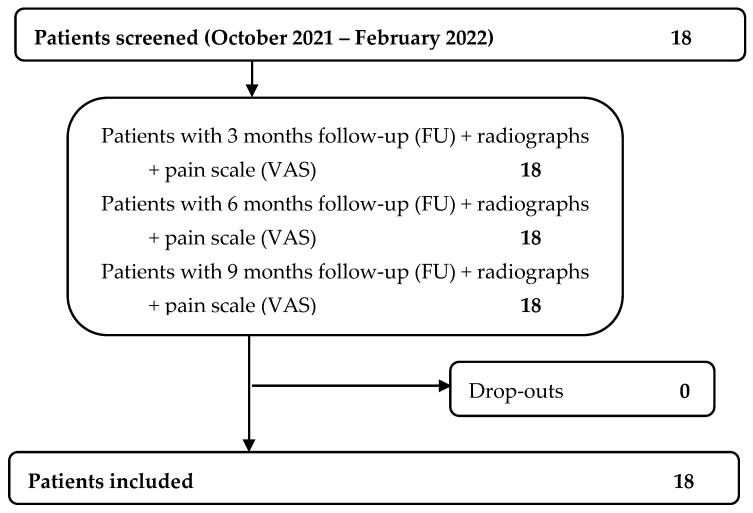
Patients’ flowchart.

**Figure 2 jpm-13-00357-f002:**
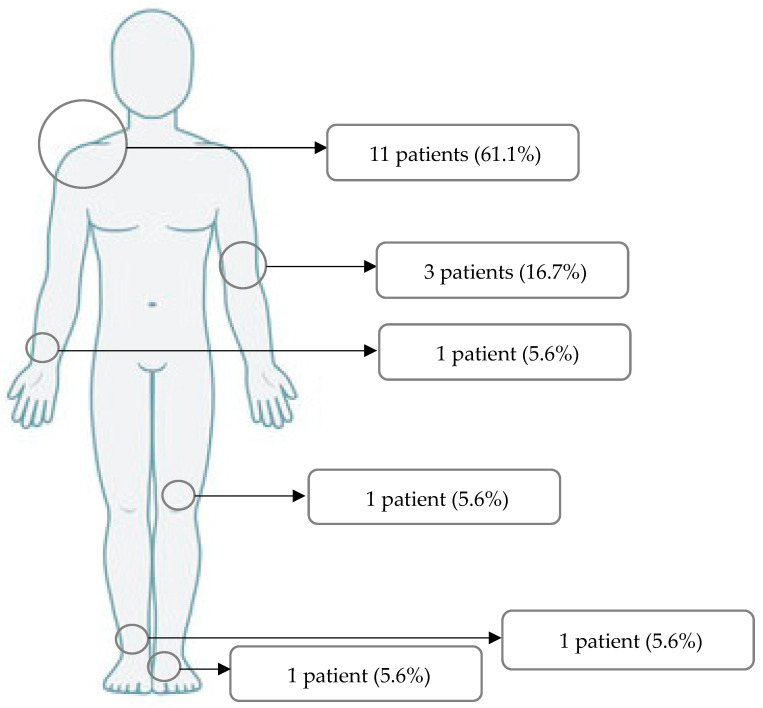
Regions of surgery.

**Figure 3 jpm-13-00357-f003:**
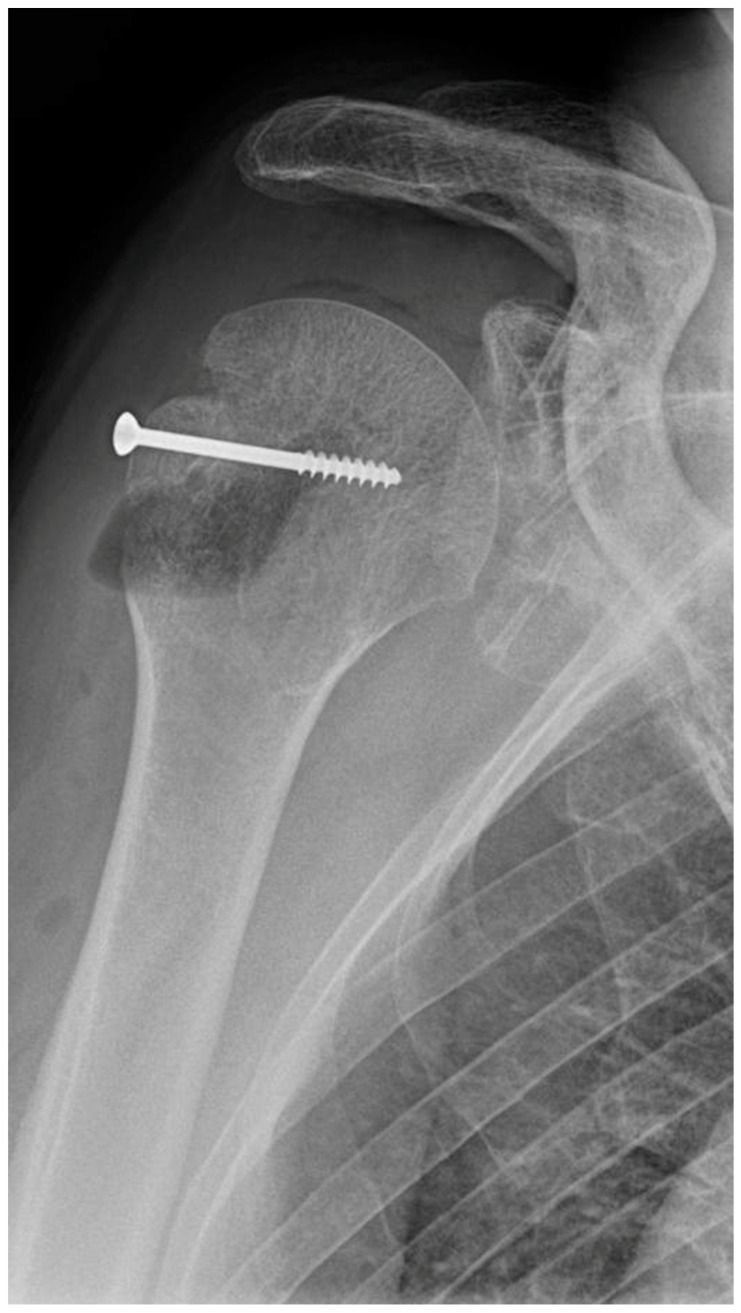
Material failure of MAGNEZIX^®^ CS screws used for osteosynthesis in a glenoid fracture after anterior shoulder dislocation.

**Table 1 jpm-13-00357-t001:** Patients’ information: BMI = body mass index, P = pain (visual analogue score), R = radiolucency (no—0, yes—1), MF = material failure (no—0, yes—1), I = infection (no—0, yes—1).

**Patient #**	**Sex (Male—0, Female—1)**	**Smoker (No—0, Yes—1)**	**Age (in Years)**	**Region of Surgery**	**BMI**	**Number of MAGNEZIX^®^ CS Screws**		**3-Month FU**					**6-Month FU**					**9-Month FU**					**Revision Surgery (No—0, Yes—1)**
								P	R	MF	I		P	R	MF	I		P	R	MF	I		
1	1	1	43	Elbow	37.0	2		0	0	0	0		0	0	0	0		0	0	0	0		0
2	0	1	39	Shoulder	33.6	2		3	1	0	0		1	1	0	0		0	0	0	0		0
3	1	0	40	Elbow	23.4	2		0	1	0	0		0	0	0	0		0	0	0	0		0
4	0	0	9	Shoulder	20.5	1		0	0	0	0		0	0	0	0		0	0	0	0		0
5	0	0	15	Forearm	21.2	1		1	0	0	0		0	0	0	0		0	0	0	0		1
6	0	0	55	Shoulder	29.0	2		0	1	0	0		0	0	0	0		0	0	0	0		0
7	0	0	42	Shoulder	28.9	1		3	0	0	1		2	0	0	0		1	0	0	0		1
8	0	1	21	Shoulder	35.9	1		0	1	0	0		0	0	0	0		0	0	0	0		0
9	0	0	25	Shoulder	24.0	2		3	0	0	0		1	0	0	0		1	0	0	0		0
10	0	1	25	Shoulder	25.1	2		2	1	1	0		0	0	1	0		0	0	1	0		0
11	0	0	24	Shoulder	25.1	2		0	0	0	0		0	0	0	0		0	0	0	0		0
12	1	0	29	Foot	18.3	2		2	0	0	0		0	0	0	0		0	0	0	0		0
13	0	1	27	Shoulder	27.4	2		3	1	0	0		6	1	0	0		4	1	1	0		0
14	0	0	12	Elbow	18.8	2		0	0	1	0		2	0	1	0		1	0	1	0		0
15	1	0	63	Ankle	27.0	2		3	1	0	1		2	1	0	0		0	0	0	0		0
16	1	0	39	Knee	22.6	2		0	1	0	0		1	1	0	0		0	1	0	0		0
17	0	0	81	Shoulder	25.5	4		2	1	1	0		1	1	1	0		1	0	1	0		0
18	0	0	64	Shoulder	24.0	2		0	1	0	0		0	0	0	0		0	0	0	0		0

## Data Availability

Not applicable.

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
