# Peer review of "High Complication Rate and High Percentage of Regressing Radiolucency in Magnesium Screw Fixation in 18 Consecutive Patients"

_jpm, 2023, doi:10.3390/jpm13020357_

Round 1

Reviewer 1 Report

- Please specify wether the data has been aquired in a prospective or retrospective way.

- A complication rate of 33% is very high - please compare it to complication rates of non degradable implants for the procedures (e.g. latarjet) and other biodegradable implants.

- The description of image one should be improved.

Author Response

  • This study is a retrospective case series.
  • The only studies that describe usable complication rates based on their case numbers are studies of hallux valgus surgery. There are no differences compared to conventional materials or other biodegradable materials. Regarding to complication rates in shoulder surgery, there are no comparative clinical studies. In the cited biomechanical study, there was no difference in material failure. Overall short term complications after latarjet procedures are indicated with 6-7% (Hurley et al.)
  • Image 1. Material failure of MAGNEZIX® CS screws used for osteosynthesis in glenoid fracture after anterior shoulder dislocation

Reviewer 2 Report

The paper contains original information on the clinical observation of 18 patients after surgical treatment of various traumatic and orthopedic pathologies between 9 and 64 years of age of both sexes. All patients underwent surgical treatment using resorbable magnesium alloy implants MAGNEZIX® CS (Syntellix AG, Hannover, Germany). The results were assessed by radiography and clinical status assessment. Complications related to the breakage of the metal construct, SSI, and the zone of bone resorption (lucidity) on radiography at 3, 6, and 9 months after implantation were taken into account. 

In the section "Materials and methods" a large range of age of patients draws attention. The fact is that regenerative abilities of children and adults are very different. Peculiarities of bone tissue metabolism and consolidation process have significant differences in childhood and adulthood. The patient sample (18 patients) is small and heterogeneous, which does not allow us to reliably assess the results and draw conclusions.

There is also a significant difference in implantation areas and types of surgical treatment. It is not correct to group surgical interventions in the shoulder and large joints of the lower extremities because of the different load on the operated structures. The consolidation of bone structures of  glenoid and femoral condyles, the ankle joint, and the distal third of the forearm is absolutely unequal. The timing of consolidation in the above parts varies.

It is reasonable to select patients taking into account their age, surgical area, comorbidity, including bone metabolic disorders, and compliance.

CONCLUSION

1. The title of the article does not reflect the essence of the work;

2. It is necessary to determine the study design;

3. It is necessary to take into account and specify the features associated with age, the area and type of surgery, etc.

Author Response

1 - The title got changed.

2 - It's a retrospective case series.

3 - Thanks for this honest statement. Due to the study design, further differentiation is not really possible. But it's definitely a big limitation of this study. Nevertheless our case series gives a solid overview concerning the usage possibilities of magnesium based screws, in particular with its focus on shoulder surgeries.

The fact, that this kind of implant did not work out very well in our hands, is necessary to present.

Round 2

Reviewer 2 Report

The article is recommended for publication with corrections made by the authors.